# Analytical Solution for Multi-Energy Groups of Neutron Diffusion Equations by a Residual Power Series Method

**Mohammed Shqair [1],\*, Ahmad El-Ajou [2]**  **and Mazen Nairat [3]**

[1]  Physics Department, Faculty of Science and Humanities, Prince Sattam bin Abdulaziz University, 11942 Al-kharj, Saudi Arabia

[2]  Department of Mathematics, Faculty of Science, Al Balqa Applied University, Salt 19117, Jordan

[3]  Department of Physics, Faculty of Science, Al Balqa Applied University, Salt 19117, Jordan

**\*** Correspondence: shqeeeer@gmail.com

**Abstract:** In this paper, a multi-energy groups of a neutron diffusion equations system is analytically solved by a residual power series method. The solution is generalized to consider three different geometries: slab, cylinder and sphere. Diffusion of two and four energy groups of neutrons is specifically analyzed through numerical calculation at certain boundary conditions. This study revels sufficient analytical description for radial flux distribution of multi-energy groups of neutron diffusion theory as well as determination of each nuclear reactor dimension in criticality case. The generated results are compatible with other different methods data. The generated results are practically efficient for neutron reactors dimension.

**Keywords:** multi-group; diffusion equation; residual power series; radial flux

---

## 1. Introduction

The nuclear reactor is a complex fuel system competition, with reflectors, coolants, control rods and other parts. The design and analysis of such reactors for different operating methods is part of a complex task involving several disciplines of nuclear engineering.

Among the most important is the determination of neutron flux distribution within the reactor core and finding the critical dimension and mass. This issue has received considerable attention in the field of reactor physics in the past decades.

The balance between neutron production from fission chain reaction and neutron loss due to radiative capture and neutron leakage must be always achieved. This balance is known as criticality, it can be mathematically represented by the steady-state neutron transport equation which can be simplified using Fick's law [1] to neutron diffusion equations. Because neutrons in a reactor have different velocities it's more convenient to represent their fluxes by multi-energy groups of neutron diffusion equations.

The nuclear reactor theory defines the probability of taking place at certain reaction by the concept of cross section which is defined as the effective size of a reacted nucleus. The cross section in nuclear reactions could be characterized due to fission, $\sigma_f$; absorption, $\sigma_a$; scattering; or $\sigma_\gamma$ radiative capture. The macroscopic cross section is a product of atomic density of the target (N) and the cross section $(\sigma) : \Sigma = N\sigma$. It is appointed to the $i$th energy group cross section as $\Sigma i$ in multi-energy groups of neutrons system, and for each nuclear reaction in the $i$th energy group its can be appointed for example $\Sigma_{f2}$ is the second energy group macroscopic fission cross section, and especially in the scattering from the ith energy group to $j$th energy group the macroscopic cross section will be in the specific form as $\Sigma sij$ and the total scattering from the $i$th energy group to the all other energy groups will be $\sum \Sigma sij$.

Neutrons of multi-energy groups move with different speeds inside the reactors. Our analysis is focused on the two and four energy groups of neutrons systems due to the necessary simplification that already applied at other alternative solution like classical methods and transport theory data [2].

The present study introduces sufficient analytical procedure based on residual power series method (RPSM) [3–12] to provide general solution to multi-energy groups of neutron diffusion equations in rectangular, cylindrical, and spherical geometries. RPSM is an effective technique to solve multi-energy groups of neutron diffusion equations without discretization, perturbation or linearization. RPSM constructs an approximate analytical solution in a practical truncated series form by using the residual error concept.

The RPSM has many characteristic features [6]; Taylor's expansion is obtained from the solution; as a result, an exact solution is available when the solution is an elementary function. Moreover, solutions and all derivatives are applicable to each arbitrary point in a determined interval. The RPSM also requires small computational requirements at high resolution and less time.

Neutron diffusion equations are solved at several cases and neutron diffusion in a hemisphere with mixed boundary conditions are presented [13]. Both homotopy analysis and Adomian decomposition methods are used to find the solution of one energy group of neutron diffusion equation in hemispherical and cylindrical reactors [14]. Moreover, the homotopy perturbation method is used to generate particular solution of one energy group of neutron diffusion equations in hemispherical symmetry [15]. An alternative method is used to solve one energy group of a neutron diffusion equation in cylindrical symmetry [16]. There are other efforts to solve different cases of neutron diffusion equations [17–21].

The present study provides general analytic approximate solution for the multi-energy groups of neutron diffusion at any particular symmetry. The proposed RPSM does not only describe neutron diffusions at different reactor geometries but it also provides associated flux distributions. The developed analytical formalism is numerically computed, the obtained result is compared to the available benchmark data generated from classical method and transport theory (*when the data is available*) to show the efficient of RPSM.

The following outlines the study: in the following section, the multi-energy groups of neutron diffusion is described analytically, and the outline of the RPSM is illustrated in Section 3. Section 4 covers how RPSM is applied in solving equations. Numerical examples are given in Section 5 to illustrate the capability of the proposed method. The study ends in Section 6 with some concluding observations.

## 2. Multi-Energy Groups of Neutron Diffusion System

The solution for multi-energy groups of neutron diffusion equations system of the following form is assumed to have a unique solution in the interval of integration:

$$
\begin{aligned}
\nabla^2 \varnothing_1(r) + C_{1,1}\varnothing_1(r) + C_{1,2}\varnothing_2(r) + C_{1,3}\varnothing_3(r) + \cdots + C_{1,n}\varnothing_n(r) = 0, \\
\nabla^2 \varnothing_2(r) + C_{2,1}\varnothing_1(r) + C_{2,2}\varnothing_2(r) + C_{2,3}\varnothing_3(r) + \cdots + C_{2,n}\varnothing_n(r) = 0, \\
\nabla^2 \varnothing_3(r) + C_{3,1}\varnothing_1(r) + C_{3,2}\varnothing_2(r) + C_{3,3}\varnothing_3(r) + \cdots + C_{3,n}\varnothing_n(r) = 0, \\
\nabla^2 \varnothing_n(r) + C_{n,1}\varnothing_1(r) + C_{n,2}\varnothing_2(r) + C_{n,3}\varnothing_3(r) + \cdots + C_{n,n}\varnothing_n(r) = 0,
\end{aligned}
\tag{1}
$$

where $C_{ii}$ is known as a group buckling, $D_i$ is a group diffusion coefficient, and $C_{ij}$ is a constant connects between fluxes in different energy groups of neutrons. Respectively $C_{ii}$, $C_{ij}$, and $D_i$ are defined as following:

$$
\begin{cases}
C_{ii} = \dfrac{\chi_i \nu_i \sum_{fi} - \left(\sum_{\gamma i} + \sum \Sigma_{\mathrm{sij}}\right)}{D_i}, \\
C_{ij} = \dfrac{\sum \Sigma_{\mathrm{sji}} + \chi_i \nu_j \sum_{fj}}{D_i}, \\
D_i = \dfrac{1}{3\left(\sum_{fi} + \Sigma_{sii} + \sum \Sigma_{\mathrm{sij}} + \Sigma_{\gamma i}\right)}.
\end{cases}
\tag{2}
$$

The constants in Equation (2) have been defined in term of different macroscopic cross sections, the number of neutrons produced per fission for each group ($\nu_i$) and the fraction of fission neutrons that emitted with energies in the $i$th group ($\chi_i$).

This system of equations describes behavior of the neutrons in the nuclear reactors where each flux $\varnothing_i$ expresses the neutron flux with specific speed. Each flux is maximum at the center of the reactor, its derivative vanishes, so the initial conditions can be written as

$$\varnothing_i(0) = h_i, \quad \varnothing_i'(0) = 0, \quad i = 1, 2, \ldots, n \tag{3}$$

where the fluxes $\varnothing_i(r)$, $i = 1, 2, \ldots, n$ are functions of independent variable $r$, and $h_i \in \mathbb{R}$. Throughout it is assumed that $\varnothing_i(r)$, $i = 1, 2, \ldots, n$ are analytic functions for $r \geq 0$.

The system of equations in Equation (1) can be simplified by taking into consideration the basic nuclear reactor theory fact which issuer that the geometrical buckling $B^2$ should be equal to the material buckling for all energy groups in the criticality case, i.e.,

$$\begin{aligned}
\nabla^2\varnothing_1(r) + B^2\varnothing_1(r) &= 0, \\
\nabla^2\varnothing_2(r) + B^2\varnothing_2(r) &= 0, \\
\nabla^2\varnothing_3(r) + B^2\varnothing_3(r) &= 0, \\
&\vdots \\
\nabla^2\varnothing_n(r) + B^2\varnothing_n(r) &= 0.
\end{aligned} \tag{4}$$

Substitution the values of $B^2$ leads to

$$\begin{aligned}
-B^2\varnothing_1(r) + C_{1,1}\varnothing_1(r) + C_{1,2}\varnothing_2(r) + C_{1,3}\varnothing_3(r) + \cdots + C_{1,n}\varnothing_n(r) &= 0, \\
-B^2\varnothing_2(r) + C_{2,1}\varnothing_1(r) + C_{2,2}\varnothing_2(r) + C_{2,3}\varnothing_3(r) + \cdots + C_{2,n}\varnothing_n(r) &= 0, \\
-B^2\varnothing_3(r) + C_{3,1}\varnothing_1(r) + C_{3,2}\varnothing_2(r) + C_{3,3}\varnothing_3(r) + \cdots + C_{3,n}\varnothing_n(r) &= 0, \\
&\vdots \\
-B^2\varnothing_n(r) + C_{n,1}\varnothing_1(r) + C_{n,2}\varnothing_2(r) + C_{n,3}\varnothing_3(r) + \cdots + C_{n,n}\varnothing_n(r) &= 0.
\end{aligned} \tag{5}$$

This system of equations shows that the ratio of each two fluxes is constant, classical solution, like Cramer's rule, determines the value of $B^2$ as well as finds each flux separately for any number of energy groups for any reactor geometry [22,23].

## 3. Basic Idea of the RPSM

The basic definition in addition to the basic theories of the RPSM and its applicability to different types of differential equations is given in [3–6]. To provide convenience to the reader, we will provide a review of the RPSM by illustrating the following algorithm:

**Step 1**: Write the $n$th-order differential equation in the following form:

$$\varnothing^{(n)}(r) = F\left[r, \varnothing, \varnothing', \varnothing'', \ldots, \varnothing^{(n-1)}\right], \tag{6}$$

subject to the initial conditions

$$\varnothing^{(i)}(r_0) = h_i, \quad i = 0, 1, 2, \ldots, n-1 \tag{7}$$

where $F$ is a function of $r, \varnothing, \varnothing', \varnothing'', \ldots, \varnothing^{(n-1)}$ and $\varnothing$ is unknown analytic function of $r$ on a neighborhood of $r_0$.

**Step 2**: Assume the solution has the following form:

$$\varnothing(r) = \sum_{m=0}^{\infty} \frac{c_m}{m!}(r - r_0)^m. \tag{8}$$

According to the initial conditions (7), the solution can be expressed in the form:

$$\varnothing(r) = \sum_{m=0}^{n-1} \frac{h_m}{m!} (r - r_0)^m + \sum_{m=n}^{\infty} \frac{c_m}{m!} (r - r_0)^m. \tag{9}$$

**Step 3**: Define the $k$th-truncated series of $\varnothing(r)$ as follows

$$\varnothing_k(r) = \sum_{m=0}^{n-1} \frac{h_m}{m!} (r - r_0)^m + \sum_{m=n}^{k} \frac{c_m}{m!} (r - r_0)^m, \quad k = n,\ n+1, n+2, \ldots . \tag{10}$$

**Step 4**: Define the $k$th-residual functions as follows

$$\mathrm{Res}^k(r) = \varnothing_k^{(n)}(r) - F\Big[t, \varnothing_k,\ \varnothing_k',\varnothing_k'', \ldots,\ \varnothing_k^{(n-1)}\Big], \quad k = n,\ n+1, n+2, \ldots . \tag{11}$$

**Step 5**: Substitute the form of $\varnothing_k(r)$ into Equation (11).
**Step 6**: Obtain the $(k-n)$th derivative of $\mathrm{Res}^k(r)$ as:

$$\frac{d^{k-n}}{dr^{k-n}} \mathrm{Res}^k(r), \quad k = n,\ n+1, n+2, \ldots . \tag{12}$$

**Step 7**: Solve the following algebraic equations for $k = n,\ n+1, n+2, \ldots$, equation by equation

$$\frac{d^{k-n}}{dr^{k-n}} \mathrm{Res}^k(r_0) = 0, \tag{13}$$

then obtain the values of the unknown coefficients; $c_n$, $c_{n+1}$, $c_{n+2}, \ldots,\ c_k$, respectively.

**Step 8**: Collect the obtained forms of $c_m$ and $\varnothing_m(r)$ for each $m = 0, 1, 2, \ldots, k$ in term of expanded series, then we obtain the $k$th-approximate solution to the Equations (6) and (7).

**Step 9**: If there are a pattern in the coefficients of the series as terms of some well-known elementary functions, then we have the exact solution $\varnothing(r)$.

## 4. Analytical Solution by the RPSM

RPSM is employed to find sufficient series solution for the multi-energy groups of neutrons time-independent neutron diffusion system of Equation (1) for three nuclear reactor essential geometries, namely spherical, cylindrical and slab reactors which will be studied respectively.

### 4.1. Multi-Energy Groups of Neutrons Spherical Reactor

The time-independent diffusion system of multi-energy groups of neutrons at spherical reactor can be written as:

$$
\begin{aligned}
r\varnothing_1''(r) + 2\varnothing_1'(r) + r(C_{1,1}\varnothing_1(r) + C_{1,2}\varnothing_2(r) + C_{1,3}\varnothing_3(r) + \cdots + C_{1,n}\varnothing_n(r)) &= 0, \\
r\varnothing_2''(r) + 2\varnothing_2'(r) + r(C_{2,1}\varnothing_1(r) + C_{2,2}\varnothing_2(r) + C_{2,3}\varnothing_3(r) + \cdots + C_{2,n}\varnothing_n(r)) &= 0, \\
r\varnothing_3''(r) + 2\varnothing_3'(r) + r(C_{3,1}\varnothing_1(r) + C_{3,2}\varnothing_2(r) + C_{3,3}\varnothing_3(r) + \cdots + C_{3,n}\varnothing_n(r)) &= 0, \\
&\ \ \vdots \\
r\varnothing_n''(r) + \varnothing_n'(r) + r(C_{n,1}\varnothing_1(r) + C_{n,2}\varnothing_2(r) + C_{n,3}\varnothing_3(r) + \cdots + C_{n,n}\varnothing_n(r)) &= 0.
\end{aligned} \tag{14}
$$

The RPSM express the solution of Equation (14) subject to the initial conditions Equation (3) as a power series expansion about the initial point $r = 0$. The series solution is supposed to take the form $\varnothing_i(r) = \sum_{m=0}^{\infty} a_{i,m} r^m$, $i = 1, 2, \ldots, n$.

Since $\varnothing_i(r), \varnothing'_i(r)$, $i = 1, 2, \ldots, n$ satisfy the initial conditions, the solutions will be $\varnothing_i(r) = h_i + \sum_{m=2}^{\infty} a_{im}r^m$, $i = 1, 2, \ldots, n$. Moreover, it could be approximately the $k$th-truncated series:

$$\varnothing_i^k(r) = h_i + \sum_{m=2}^{k} a_{im}r^m, \quad i = 1, 2 \ldots, n. \tag{15}$$

The coefficients $a_{i,m}$, $i = 1, 2, \ldots, n$, $m = 2, 3, \ldots, k$, are determined by defining residual functions and the $k$th-residual functions, respectively, as follows:

$$\text{Res}_i(r) = r\varnothing''_i(r) + 2\varnothing'_i(r) + r\sum_{j=1}^{n} C_{ij}\varnothing_j(r), \ i = 1, 2, \ldots, n \tag{16}$$

$$\text{Res}_i^k(r) = r\frac{d^2\varnothing_i^k(r)}{dr^2} + 2\frac{d\varnothing_i^k(r)}{dr} + r\sum_{j=1}^{n} C_{ij}\varnothing_j^k(r), \ i = 1, 2, \ldots, n \tag{17}$$

Substitute Equation (15) into Equation (17), we have:

$$\begin{aligned}
\text{Res}_i^k(r) = &\sum_{m=2}^{k} m(m-1)a_{im}r^{m-1} + 2\sum_{m=2}^{k} ma_{im}r^{m-1} \\
&+ \sum_{j=1}^{n} C_{ij}\left(h_j r + \sum_{m=2}^{k} a_{jm}r^{m+1}\right), \ i = 1, 2, \ldots, n
\end{aligned} \tag{18}$$

It is clear that, $\text{Res}_i(r) = \lim_{k\to\infty} \text{Res}_i^k(r)$, $i = 1, 2, \ldots, n$. Moreover, $\text{Res}_i(r) = 0$, $i = 1, 2, \ldots, n$ for $r \geq 0$. This show that $\text{Res}_i(r)$, $i = 1, 2, \ldots, n$ is infinitely many times differentiable at $r = 0$. On the other hand, $\frac{d^{k-1}}{dr^{k-1}}\text{Res}_i(0) = \frac{d^{k-1}}{dr^{k-1}}\text{Res}_i^k(0) = 0$, this fact is a basic rule in RPSM and its applications.

Now, to obtain second approximate solutions, $k = 2$ is placed in Equation (18), differentiate both sides with respect to $r$ and substitute $r = 0$, to conclude

$$\frac{d}{dr}\text{Res}_i^2(r) = 2a_{i2} + 4a_{i2} + \sum_{j=1}^{n} C_{ij}h_j + \left(\sum_{j=1}^{n} 3C_{ij}a_{j2}\right)r^2, \ i = 1, 2, \ldots, n. \tag{19}$$

The fact that $\frac{d^{k-1}}{dr^{k-1}}\text{Res}_i^k(0) = 0$, $i = 1, 2, \ldots, n$ gives the following values for $a_{i2}$:

$$a_{i2} = \frac{-1}{2*3}\sum_{j=1}^{n} C_{ij}h_j, \ i = 1, 2, \ldots, n \tag{20}$$

Thus, the second truncated series solutions for Equation (14) can be written as

$$\varnothing_i^2(r) = h_i - \frac{1}{2*3}\left(\sum_{j=1}^{n} C_{ij}h_j\right)r^2, \ i = 1, 2, \ldots, n$$

Similarly, to find the third approximate solution, $k = 3$ is placed in Equation (18), differentiate both sides twice with respect to $r$ with substituting $r = 0$ and use the fact $\frac{d^{k-1}}{dr^{k-1}}\text{Res}_i^k(0) = 0$, $i = 1, 2, \ldots, n$, then we get that $a_{i3} = 0$, $i = 1, 2, \ldots, n$. Hence, the 3rd-truncated series is the same as the 2nd-truncated series.

Repeat same procedure for $k = 4$, $k = 5$ and $k = 6$, values of $a_{i4}$, $a_{i5}$ and $a_{i6}$ are given in the following formulas:

$$a_{i4} = \frac{1}{2*3*4*5}\sum_{l=1}^{n} C_{il} \sum_{j=1}^{n} C_{lj}h_j, \ i = 1, 2, \ldots, n,$$

$a_{i5} = 0$, $i = 1, 2, \ldots, n$,

$$a_{i6} = \frac{-1}{2*3*4*5*6*7} \sum_{r=1}^{n} C_{ir} \sum_{l=1}^{n} C_{rl} \sum_{j=1}^{n} C_{lj}h_j, \ i = 1, 2, \ldots, n. \tag{21}$$

Define the recurrence relation: $T_{i,0} = h_i$, and $T_{i,m} = \sum_{j=1}^{n} C_{ij}T_{j,(m-2)}$, $m = 2, 4, 6, \ldots$, then

$$\begin{aligned} a_{i0} &= T_{i,0}, \\ a_{i2} &= \tfrac{-1}{3!}T_{i,2}, \\ a_{i4} &= \tfrac{1}{5!}T_{i,4}, \ i = 1, 2, \ldots, n, \\ a_{i6} &= \tfrac{-1}{7!}T_{i,6}, \ i = 1, 2, \ldots, n. \end{aligned} \tag{22}$$

Therefore, the solution of the system (14) subject to the initial conditions (3) can be arranged in the form

$$\varnothing_i(r) = \sum_{k=0}^{\infty} (-1)^k \frac{1}{(2k+1)!} T_{i,2k} r^{2k}, \ i = 1, 2, \ldots, n. \tag{23}$$

In a special case, if $T_{i,2k} = 1$, then

$$\varnothing_i(r) = \frac{\sin r}{r}. \tag{24}$$

It should be noted that the simplification of Equation (24) coincides with a one energy group of a neutron diffusion system.

### 4.2. Multi-Energy Groups of Neutrons Cylindrical Reactor

The cylindrical reactor is considered by solving multi-energy groups of neutrons time-independent neutron diffusion as follow:

$$\begin{aligned} r\varnothing_1''(r) + \varnothing_1'(r) + r(C_{1,1}\varnothing_1(r) + C_{1,2}\varnothing_2(r) + C_{1,3}\varnothing_3(r) + \cdots + C_{1,n}\varnothing_n(r)) &= 0, \\ r\varnothing_2''(r) + \varnothing_2'(r) + r(C_{2,1}\varnothing_1(r) + C_{2,2}\varnothing_2(r) + C_{2,3}\varnothing_3(r) + \cdots + C_{2,n}\varnothing_n(r)) &= 0, \\ r\varnothing_3''(r) + \varnothing_3'(r) + r(C_{3,1}\varnothing_1(r) + C_{3,2}\varnothing_2(r) + C_{3,3}\varnothing_3(r) + \cdots + C_{3,n}\varnothing_n(r)) &= 0, \\ &\vdots \\ r\varnothing_n''(r) + \varnothing_n'(r) + r(C_{n,1}\varnothing_1(r) + C_{n,2}\varnothing_2(r) + C_{n,3}\varnothing_3(r) + \cdots + C_{n,n}\varnothing_n(r)) &= 0. \end{aligned} \tag{25}$$

Using the same methodology of RPSM which used in treating the spherical reactor in the last subsection we reach to:

$$\mathrm{Res}_i^k(r) = r\frac{d^2\varnothing_i^k(r)}{dr^2} + \frac{d\varnothing_i^k(r)}{dr} + r\sum_{j=1}^{n} C_{ij}\varnothing_j^k(r), \ i = 1, 2, \ldots, n. \tag{26}$$

Substitute Equation (15) into Equation (26), we have:

$$\mathrm{Res}_i^k(r) = \sum_{m=2}^{k} m(m-1)a_{im}r^{m-1} + \sum_{m=2}^{k} ma_{im}r^{m-1} + \sum_{j=1}^{n} C_{ij}\left(h_j r + \sum_{m=2}^{k} a_{jm}r^{m+1}\right), \ i = 1, 2, \ldots, n \tag{27}$$

To obtain the second approximate solutions, we place $k = 2$ in Equation (27), differentiate both sides of the equation with respect to $r$ and substitute $r = 0$ in the resulting equation, we conclude

$$\frac{d}{dr}\mathrm{Res}_i^2(r) = 2a_{i2} + 4a_{i2} + \sum_{j=1}^{n} C_{ij}h_j + \left(\sum_{j=1}^{n} 3C_{ij}a_{j2}\right)r^2, \ i = 1, 2, \ldots, n. \tag{28}$$

The algebraic equations, $\frac{d}{dr}\text{Res}_i^2(0) = 0$, $i = 1, 2, \ldots, n$, give the following values for $a_{i2}$:

$$a_{i2} = \frac{-1}{2*2}\sum_{j=1}^{n} C_{ij}h_j, \ i = 1, 2, \ldots, n. \tag{29}$$

Thus, the second truncated series solutions (the second approximate solutions) for Equation (25) can be written as

$$\varnothing_i^2(r) = h_i - \frac{1}{2*2}\left(\sum_{j=1}^{n} C_{ij}h_j\right)r^2, \ i = 1, 2, \ldots, n$$

Similarly, to find the third approximate solution, put $k = 3$ in Equation (27) and solve the algebraic equations, $\frac{d^2}{dr^2}\text{Res}_i^2(0) = 0$, $i = 1, 2, \ldots, n$, that give us $a_{i3} = 0$, $i = 1, 2, \ldots, n$.

Repeat same procedure for $k = 4$, $k = 5$ and $k = 6$, the values of $a_{i4}$, $a_{i5}$ and $a_{i6}$ will become available as follows:

$$a_{i4} = \frac{1}{2*2*4*4}\sum_{l=1}^{n} C_{il}\sum_{j=1}^{n} C_{lj}h_j, \ i = 1, 2, \ldots, n,$$
$$a_{i5} = 0, \ i = 1, 2, \ldots, n, \tag{30}$$
$$a_{i6} = \frac{-1}{2*2*4*4*6*6}\sum_{r=1}^{n} C_{ir}\sum_{l=1}^{n} C_{rl}\sum_{j=1}^{n} C_{lj}h_j, \ i = 1, 2, \ldots, n.$$

Using of the recurrence relation: $T_{i,0} = h_i$, and $T_{i,m} = \sum_{j=1}^{n} C_{ij}T_{j,(m-2)}$, $m = 2, 4, 6, \ldots$, will help us write the series coefficients obtained as follows:

$$a_{i0} = T_{i,0}$$
$$a_{i2} = \frac{-1}{2*2}T_{i,2}$$
$$a_{i4} = \frac{1}{2*2*4*4}T_{i,4}, \ i = 1, 2, \ldots, n, \tag{31}$$
$$a_{i6} = \frac{-1}{2*2*4*4*6*6}T_{i,6}, \ i = 1, 2, \ldots, n.$$

Thus, the solution of the cylindrical reactor system (25) with the conditions (3) can be expressed in the following form:

$$\varnothing_i(r) = \sum_{k=0}^{\infty}(-1)^k\frac{1}{4^k(k!)^2}T_{i,2k}r^{2k}, \ i = 1, 2, \ldots, n. \tag{32}$$

In a special case, if $T_{i,2k} = 1$, then

$$\varnothing_i(r) = J_0(r) \tag{33}$$

where $J_0(r)$ is a Bessel's function of the first kind.

It is noted that the solution given in of Equation (33), is the same as the solution of one energy group of a neutron diffusion system.

### 4.3. Multi-Energy Groups Slab Reactor

The final example is the slab reactor, the multi-energy groups of time-independent neutron diffusion system of a slab reactor can be expressed as:

$$\varnothing_1''(r) + (C_{1,1}\varnothing_1(r) + C_{1,2}\varnothing_2(r) + C_{1,3}\varnothing_3(r) + \cdots + C_{1,n}\varnothing_n(r)) = 0,$$
$$\varnothing_2''(r) + (C_{2,1}\varnothing_1(r) + C_{2,2}\varnothing_2(r) + C_{2,3}\varnothing_3(r) + \cdots + C_{2,n}\varnothing_n(r)) = 0,$$
$$\varnothing_3''(r) + (C_{3,1}\varnothing_1(r) + C_{3,2}\varnothing_2(r) + C_{3,3}\varnothing_3(r) + \cdots + C_{3,n}\varnothing_n(r)) = 0, \tag{34}$$
$$\vdots$$
$$\varnothing_n''(r) + (C_{n,1}\varnothing_1(r) + C_{n,2}\varnothing_2(r) + C_{n,3}\varnothing_3(r) + \cdots + C_{n,n}\varnothing_n(r)) = 0.$$

According to Equation (11), the $k$th-residual function of the Equation (34) will be as follows:

$$\text{Res}_i^k(r) = \frac{d^2\varnothing_i^k(r)}{dr^2} + \sum_{j=1}^{n} C_{ij}\varnothing_j^k(r), \ i = 1, 2, \ldots, n. \tag{35}$$

Substitute Equation (15) into Equation (35), we have:

$$\text{Res}_i^k(r) = \sum_{m=2}^{k} m(m-1)a_{im}r^{m-1} + \sum_{j=1}^{n} C_{ij}\left(h_j r + \sum_{m=2}^{k} a_{jm}r^{m+1}\right), \ i = 1, 2, \ldots, n. \tag{36}$$

For, $k = 2$, differentiate both sides of Equation (36) with respect to $r$ and substitute $r = 0$, to obtain

$$\frac{d}{dr}\text{Res}_i^2(r) = 2a_{i2} + \sum_{j=1}^{n} C_{ij}h_j + \left(\sum_{j=1}^{n} 3C_{ij}a_{j2}\right)r^2, \ i = 1, 2, \ldots, n. \tag{37}$$

The solution of the algebraic equations, $\frac{d}{dr}\text{Res}_i^2(0) = 0$, $i = 1, 2, \ldots, n$ gives the following values for $a_{i2}$:

$$a_{i2} = \frac{-1}{2}\sum_{j=1}^{n} C_{ij}h_j, \ i = 1, 2, \ldots, n \tag{38}$$

Thus, the second truncated series solutions for Equation (35) can be written as

$$\varnothing_i^2(r) = h_i - \frac{1}{2}\left(\sum_{j=1}^{n} C_{ij}h_j\right)r^2, \ i = 1, 2, \ldots, n.$$

Like previous cases, the third approximate solution of the system in Equation (38) is the same as the second approximate solution since the fourth coefficient of the series (15), $a_{i3}$, is zero.

Repeat same procedure for $k = 4$, $k = 5$, and $k = 6$, values of $a_{i4}$, $a_{i5}$, and $a_{i6}$ are given in the following formulas:

$$
\begin{aligned}
a_{i4} &= \frac{1}{2*3*4}\sum_{l=1}^{n} C_{il}\sum_{j=1}^{n} C_{lj}h_j, \ i = 1, 2, \ldots, n, \\
a_{i5} &= 0, \ i = 1, 2, \ldots, n, \\
a_{i6} &= \frac{-1}{2*3*4*5*6}\sum_{r=1}^{n} C_{ir}\sum_{l=1}^{n} C_{rk}\sum_{j=1}^{n} C_{kj}h_j, \ i = 1, 2, \ldots, n.
\end{aligned}
\tag{39}
$$

Again, by the recurrence relation: $T_{i,0} = h_i$, and $T_{i,m} = \sum_{j=1}^{n} C_{ij}T_{j,(m-2)}$, $m = 2, 4, 6, \ldots$, we can arrange the coefficients of the series as follows:

$$
\begin{aligned}
a_{i0} &= T_{i,0} \\
a_{i2} &= \frac{-1}{2}T_{i,2} \\
a_{i4} &= \frac{1}{4!}T_{i,4}, \quad i = 1, 2, \ldots, n, \\
a_{i6} &= \frac{-1}{6!}T_{i,6}, \quad i = 1, 2, \ldots, n.
\end{aligned}
\tag{40}
$$

Therefore, the solution of the slab reactor system (34) with the conditions (3) can be expressed in the following form:

$$\varnothing_i(r) = \sum_{k=0}^{\infty}(-1)^k\frac{1}{(2k)!}T_{i,2k}r^{2k}, \ i = 1, 2, \ldots, n. \tag{41}$$

In a special case, if $T_{i,2k} = 1$, then $\varnothing_i(r) = \cos r$ which coincides with one energy group of a neutron diffusion system as well.

## 5. Practical Numerical Results

Generating numerical results requires specifying boundary conditions. The flux is assumed to vanish at the surface of the reactor according zero flux boundary condition. Indeed, extrapolated boundary condition (EBC) assumes the flux vanishes at small distance beyond the surface. Couple boundary conditions are considered in such a way diffusion flux vanishes at the critical dimension. As in the same procedure that used in previous works like [20,21], one of the fluxes is normalized (to one) at the core of the reactor, and the other fluxes will be calculated depending on Equation (5) as ratios of the normalized flux, for two energy groups of neutrons the ratio between thermal and fast fluxes is given in the data source [2] which convenient with Equation (5) calculations.

Following numerical examples results are compare with the classical solutions, some of them are also compared with transport theory data in case transport theory data is available.

It should be mentioned that, as any series solution, increasing the order of approximation increases the accuracy of the solution. Therefore, to obtain accurate numerical solutions, all numerical values and graphs are calculated for the 10th approximation of the power series solution obtained in Equations (23), (32), and (41) and it is compared with the 10th approximation of the series solution obtained by the classical method.

### 5.1. Two Energy Groups of Neutrons Numerical Example

The two energy groups of a neutron diffusion equations system is discussed in this section at three cases; spherical reactor, infinite cylindrical reactor and slab reactor such diffusion system satisfies the following form:

$$
\begin{aligned}
\nabla^2 \varnothing_1(r) + C_{1,1}\varnothing_1(r) + C_{1,2}\varnothing_2(r) = 0, \\
\nabla^2 \varnothing_2(r) + C_{2,1}\varnothing_1(r) + C_{2,2}\varnothing_2(r) = 0,
\end{aligned}
\tag{42}
$$

subject to the initial conditions

$$
\varnothing_i) = I, \ \varnothing_i'(0) = 0, \quad i = 1, 2.
\tag{43}
$$

Equation (42) consider only couple speeds of neutrons: fast and thermal, this simplification is previous considered at existing transport data [2].

So, the fast and thermal fluxes for spherical reactor according to RPSM formula are given by

$$
\varnothing_i(r) = \sum_{k=0}^{\infty} (-1)^k \frac{1}{(2k+1)!} T_{i,2k} r^{2k}, \ i = 1, 2,
\tag{44}
$$

and the fluxes of the cylindrical reactor are given by

$$
\varnothing_i(r) = \sum_{k=0}^{\infty} (-1)^k \frac{1}{4^k (k!)^2} T_{i,2k} r^{2k}, \ i = 1, 2,
\tag{45}
$$

whereas for slab reactor is

$$
\varnothing_i(r) = \sum_{k=0}^{\infty} (-1)^k \frac{1}{(2k)!} T_{i,2k} r^{2k}, \ i = 1, 2,
\tag{46}
$$

The derived analytical formalism is computationally determined not only to verify the theory but also to compare numerical results with a classical solution [22–25]. For this purpose, the Mathematics 9 software package is implemented to generate numerical solutions. The solution is obtained numerically for cross sections related to interactions of fast and thermal neutron diffusing in 93% enriched Uranium for geometries slab, sphere and cylinder geometries. Table 1 obtained from [2] is correspondingly used.

<div align="center">**Table 1.** Two Group Data.</div>

| Fast Energy Group | | |
|---|---|---|
| $\Sigma_{f1} = 0.0010484$ cm$^{-1}$ $\Sigma_{S12} = 0.029227$ cm$^{-1}$ | $\Sigma_{\gamma1} = 0.0010046$ cm$^{-1}$ $\nu_1 = 2.5$ | $\Sigma_{S11} = 0.62568$ cm$^{-1}$ $\chi_1 = 1.0$ |
| Thermal Energy Group | | |
| $\Sigma_{f2} = 0.050632$ cm$^{-1}$ $\Sigma_{S21} = 0.0000$ cm$^{-1}$ | $\Sigma_{\gamma2} = 0.025788$ cm$^{-1}$ $\nu_2 = 2.5$ | $\Sigma_{S22} = 2.44383$ cm$^{-1}$ $\chi_2 = 0.0$ |

According this data, values of $C_{ij}$, $i, j = 1, 2$ are determined as shown in Table 2.

<div align="center">**Table 2.** The values of the coefficients $C_{ij}$, $i, j = 1, 2$ are calculated from Equation (3).</div>

| | | | |
|---|---|---|---|
| $C_{11} = -0.0564834$ | $C_{12} = 0.249474$ | $C_{21} = 0.220978$ | $C_{22} = -0.577793$ |

Following subsections show numerical results for each reactor geometry.

### 5.1.1. Spherical Reactor

The spherical reactor is studied by finding its critical radius of a 93% enriched Uranium and distribution of both fast and thermal fluxes as well as their sum, the total flux.

To calculate the spherical reactor critical radius $a_c$, Equation (44) is used, at ZF and EBC boundary conditions, respectively. The generated data are listed in Table 3, other numerical data that generated by classical methods and transport theory is illustrated for comparison.

<div align="center">**Table 3.** The critical radius $a_c$ of a 93 % enriched Uranium spherical reactor.</div>

| BC | Classical Method | RPSM | Transport Theory |
|---|---|---|---|
| ZF | 17.120 | 17.120 | - |
| EBC | 16.251 | 16.251 | 16.049836 |

This reactor critical radius (using RPSM) reproduced that of classical Method and it is in good approximation with it and with our benchmark (Transport Theory).

The fluxes distribution in the spherical reactor is reported in Table 4 and plotted in Figure 1. Table 4 provides the normalized thermal, fast and total flux values across the system for different values of $r/a_c$.

<div align="center">**Table 4.** The normalized thermal, fast and total fluxes in a spherical geometry.</div>

| Flux | Method | $r/a_c$ | 0.0 | 0.25 | 0.50 | 0.75 | 1.0 |
|---|---|---|---|---|---|---|---|
| Fast | Classical | | 2.7671 | 2.5238 | 1.7802 | 1.0063 | 0.1835 |
| | RPSM | | 2.7671 | 2.5238 | 1.7802 | 1.0063 | 0.1835 |
| Thermal | Classical | | 1.0000 | 0.9121 | 0.6759 | 0.3637 | 0.0663 |
| | RPSM | | 1.0000 | 0.9121 | 0.6759 | 0.3637 | 0.0663 |
| Total | Classical | | 3.7671 | 3.4359 | 2.5460 | 1.3699 | 0.2498 |
| | RPSM | | 3.7671 | 3.4359 | 2.5460 | 1.3699 | 0.2498 |

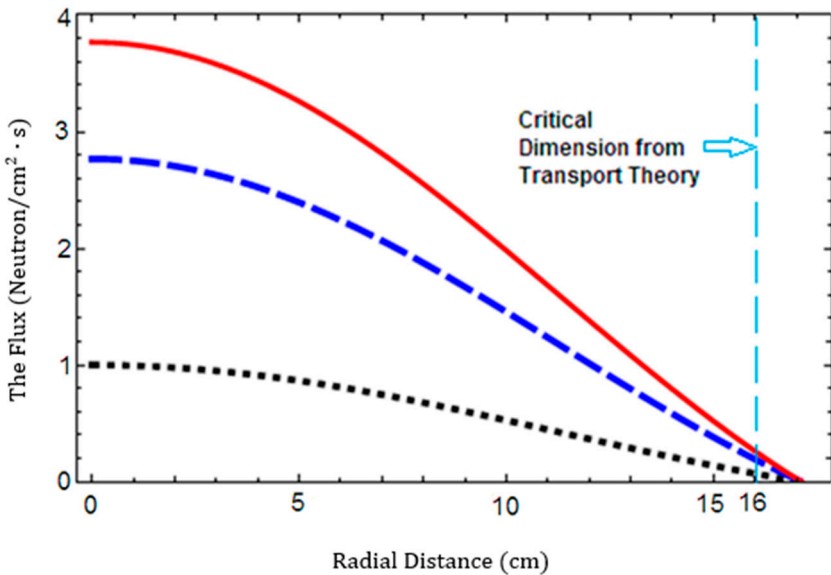

**Figure 1.** Two energy groups fluxes for a sphere made of 93% enriched U. Dotted line: Thermal flux, Dashed line: Fast flux, Solid line: Total fluxes.

Table 4 and Figure 1 gives the values of the sub fluxes and their total, it is obvious that all fluxes converge at the same point.

The RPSM values are compatible with those obtained from classical calculations [22–25] and transport data for critical radius [2].

The sufficient of RPSM solution for two energy groups of neutrons system is evident, the method can definitely reproduce canonical results for such geometry.

### 5.1.2. Infinite Cylindrical Reactor

RPSM is used in this section to calculate the reactor critical radius $a_c$ of 93% enriched Uranium cylindrical reactor, Equation (45) is implemented with ZF and EBC boundary conditions, the results are listed in Table 5.

**Table 5.** The cylindrical reactor critical radius for two boundary conditions

| BC | Classical Method | RPSM |
|---|---|---|
| ZF | 13.105 | 13.105 |
| EBC | 12.236 | 12.236 |

The flux distribution of such cylindrical reactor is reported in Table 5 at which fluxes values across the system for different values of $r/a_c$ are listed. The RPSM values are compared with classical diffusion calculations [22–25]. Transport results for both nuclear reactor critical radius and fluxes distribution are unfortunately not available to compare with. Figure 2 illustrates graphically the fluxes distribution in this fissile system, while the numerical values of them is tabulated in Table 6, the maximum value of the flux is at the axis of the cylinder ($r = 0$), it decreases towards the surface as implied by the symmetry of the system.

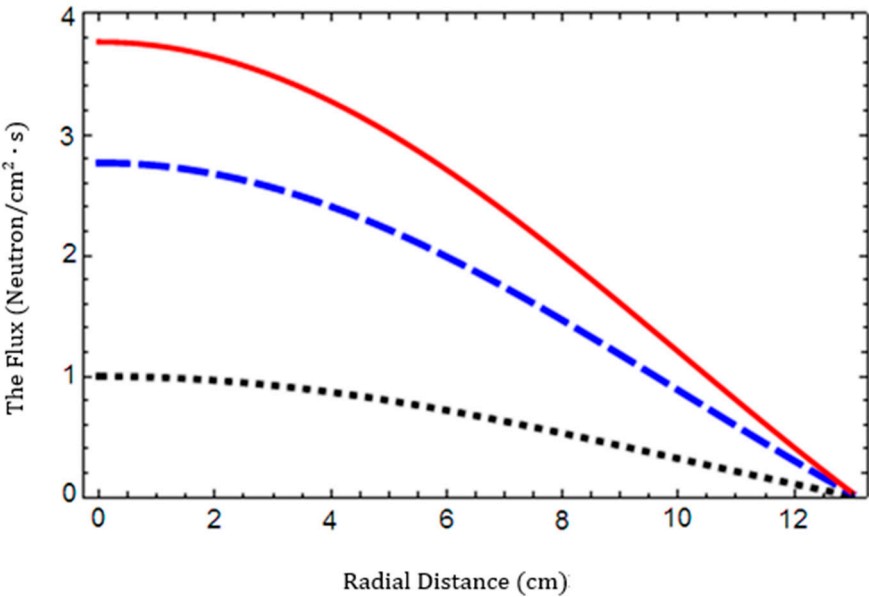

**Figure 2.** Two energy groups fluxes of a cylinder made of 93% enriched U. Dotted line: Thermal flux, Dashed line: Fast flux, Solid line: Total fluxes.

**Table 6.** The normalized thermal, fast, and total fluxes values across a cylindrical geometry.

| Flux | Method | $r/a_c$ | 0.0 | 0.25 | 0.50 | 0.75 | 1.0 |
|---|---|---|---|---|---|---|---|
| Fast | Classical | | 2.7671 | 2.5534 | 1.9615 | 1.1268 | 0.2360 |
| | RPSM | | 2.7671 | 2.5534 | 1.9615 | 1.1268 | 0.2360 |
| Thermal | Classical | | 1.0000 | 0.9228 | 0.7089 | 0.4072 | 0.0853 |
| | RPSM | | 1.0000 | 0.9228 | 0.7089 | 0.4072 | 0.0853 |
| Total | Classical | | 3.7671 | 3.4761 | 2.6704 | 1.5340 | 0.3213 |
| | RPSM | | 3.7671 | 3.4761 | 2.6704 | 1.5340 | 0.3213 |

Applying RPSM is also sufficient for two energy groups of a neutron diffusion system at cylindrical symmetry as shown in Figure 2 and Table 6.

### 5.1.3. Slab Reactor

The last two energy groups of neutrons computational analysis in this study is an infinite slab geometry. To calculate the nuclear reactor critical dimension $a_c$ of a 93% enriched Uranium slab, Equation (46) computationally implemented at ZF and EBC boundary conditions, the exact nuclear reactor critical dimension results of both RPSM and Classical Method compared with transport theory data is illustrated in Table 7.

**Table 7.** The slab reactor critical radius for two boundary conditions.

| BC | Classical Method | RPSM | Transport Theory |
|---|---|---|---|
| ZF | 8.560 | 8.560 | – |
| EBC | 7.874 | 7.874 | 7.567 |

The fluxes distribution in the slab reactor is reported in Table 8 at which fluxes values across the system for different values of $r/a_c$ are listed, the transport theory fluxes data is available to compare with, the good agreement with it can be considered as one step forward.

**Table 8.** The normalized thermal, fast and total fluxes values across a slab geometry.

| Flux | Method | $r/a_c$ | 0.0 | 0.2414 | 0.5029 | 0.74430 | 1.0 |
|------|--------|---------|-----|--------|--------|---------|-----|
| Fast | Transport | | 2.6147 | 2.4666 | 1.9923 | 1.3178 | 0.3859 |
| | Classical | | 2.7671 | 2.6130 | 2.1192 | 1.4158 | 0.5014 |
| | RPSM | | 2.7671 | 2.6130 | 2.1192 | 1.4158 | 0.5014 |
| Thermal | Transport | | 1.0000 | 0.8893 | 0.7140 | 0.4546 | 0.0555 |
| | Classical | | 1.0000 | 0.9443 | 0.7659 | 0.5117 | 0.1812 |
| | RPSM | | 1.0000 | 0.9443 | 0.7659 | 0.5117 | 0.1812 |
| Total | Transport | | 3.6147 | 3.3559 | 2.7063 | 1.7724 | 0.4414 |
| | Classical | | 3.7671 | 3.5574 | 2.8851 | 1.9275 | 0.6826 |
| | RPSM | | 3.7671 | 3.5574 | 2.8851 | 1.9275 | 0.6826 |

Figure 3 illustrates the flux distribution of a slab fissile material, the RPSM values are compared to those obtained from classical diffusion calculations as well as from transport theory.

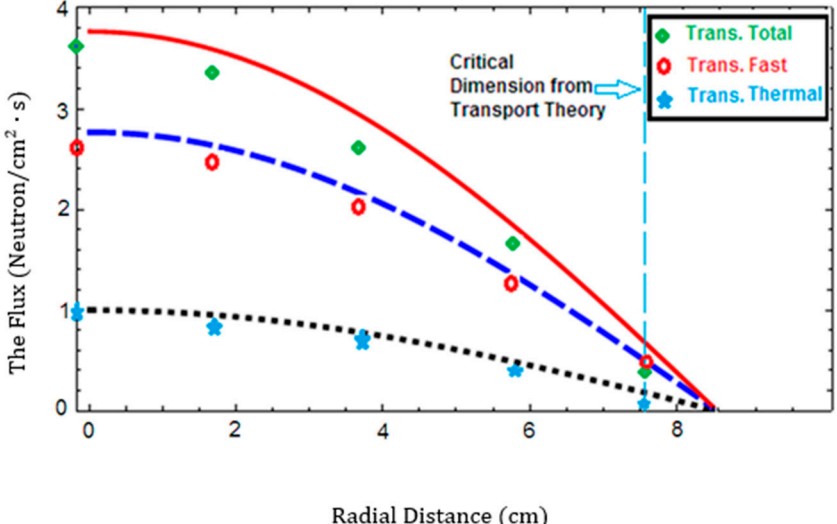

**Figure 3.** Two Energy Groups fluxes for an infinite Slab of 93% enriched U. Dotted line: Thermal flux, Dashed line: Fast flux, Solid line: Total fluxes.

*5.2. Multi-Energy Groups of Neutrons Numerical Example*

Four energy groups of a neutron diffusion equations case is discussed as one step forward, it is represented by the following system:

$$\begin{aligned}
\nabla^2 \varnothing_1(r) + C_{1,1}\varnothing_1(r) + C_{1,2}\varnothing_2(r) + C_{1,3}\varnothing_3(r) + C_{1,4}\varnothing_4(r) &= 0, \\
\nabla^2 \varnothing_2(r) + C_{2,1}\varnothing_1(r) + C_{2,2}\varnothing_2(r) + C_{2,3}\varnothing_3(r) + C_{2,4}\varnothing_4(r) &= 0, \\
\nabla^2 \varnothing_3(r) + C_{3,1}\varnothing_1(r) + C_{3,2}\varnothing_2(r) + C_{3,3}\varnothing_3(r) + C_{3,4}\varnothing_4(r) &= 0, \\
\nabla^2 \varnothing_4(r) + C_{4,1}\varnothing_1(r) + C_{4,2}\varnothing_2(r) + C_{4,3}\varnothing_3(r) + C_{4,4}\varnothing_4(r) &= 0,
\end{aligned} \tag{47}$$

subject to the initial conditions:

$$\varnothing_i(0) = h_i, \ \varnothing_i'(0) = 0, \quad i = 1, 2, 3, 4. \tag{48}$$

The spherical reactor, infinite cylindrical reactor and slab reactor are considered respectively, in addition to assuming four different speeds of neutrons.

So, the solution of spherical reactor for four energy groups of neutrons according to RPSM formula are given by

$$\varnothing_i(r) = \sum_{k=0}^{\infty} (-1)^k \frac{1}{(2k+1)!} T_{i,2k} r^{2k}, \; i = 1, 2, 3, 4, \tag{49}$$

and the fluxes of the cylindrical reactor are given by

$$\varnothing_i(r) = \sum_{k=0}^{\infty} (-1)^k \frac{1}{4^k (k!)^2} T_{i,2k} r^{2k}, \; i = 1, 2, 3, 4, \tag{50}$$

where that for slab reactor is

$$\varnothing_i(r) = \sum_{k=0}^{\infty} (-1)^k \frac{1}{(2k)!} T_{i,2k} r^{2k}, \; i = 1, 2, 3, 4. \tag{51}$$

The derived analytical formalism is computationally determined not only to verify the theory, but also to compare numerical results classical solution [22–25], Mathematics 9 software package is implemented to generate numerical solutions. The solution is obtained numerically for cross sections related to interactions of four energy groups of neutrons for slab, sphere and cylinder geometries. The following data, Table 9, obtained from [25], is used correspondingly.

**Table 9.** Group data.

| Group 1 (1.35 Mev – 10 Mev) | | |
|---|---|---|
| $\nu_1 \Sigma_{f1} = 0.0096 \text{ cm}^{-1}$ | $\Sigma_a = 0.0049 \text{ cm}^{-1}$ | $\Sigma_{S12} = 0.0831 \text{ cm}^{-1}$ |
| $\Sigma_{S13} = 0.00$ | $\Sigma_{S14} = 0.00$ | $D_1 = 2.162 \text{ cm}$ |
| | $\chi_1 = 0.575$ | |
| **Group 2 (9.1 kev - 1.35 Mev)** | | |
| $\nu_2 \Sigma_{f2} = 0.0012 \text{ cm}^{-1}$ | $\Sigma_{a2} = 0.0028 \text{ cm}^{-1}$ | $\Sigma_{S21} = 0.00 \text{ cm}^{-1}$ |
| $\Sigma_{S23} = 0.0.0585 \text{ cm}^{-1}$ | $\Sigma_{S24} = 0.00 \text{ cm}^{-1}$ | $D_2 = 1.087 \text{ cm}$ |
| | $\chi_2 = 0.425$ | |
| **Group 3 (0.4 ev – 9.1 kev)** | | |
| $\nu_3 \Sigma_{f3} = 0.0.0177 \text{ cm}^{-1}$ | $\Sigma_{a3} = 0.0.0305 \text{ cm}^{-1}$ | $\Sigma_{S31} = 0.00 \text{ cm}^{-1}$ |
| $\Sigma_{S32} = 0.00 \text{cm}^{-1}$ | $\Sigma_{S34} = 0.0651 \text{ cm}^{-1}$ | $D_3 = 0.632 \text{ cm}$ |
| | $\chi_3 = 0.0$ | |
| **Group 4 (0.0 ev – 0.4 ev)** | | |
| $\nu_4 \Sigma_{f4} = 0.1851 \text{ cm}^{-1}$ | $\Sigma_{a4} = 0.1210 \text{ cm}^{-1}$ | $\Sigma_{S41} = 0.00 \text{ cm}^{-1}$ |
| $\Sigma_{S42} = 0.00 \text{cm}^{-1}$ | $\Sigma_{S43} = 0.00 \text{cm}^{-1}$ | $D_4 = 0.354 \text{ cm}$ |
| | $\chi_4 = 0.0$ | |

values of $C_{ij}, \; i, j = 1, 2, 3, 4$ are calculated from Equation (3) and are shown in Table 10.

**Table 10.** The values of the coefficients $C_{ij}, \; i, j = 1, 2, 3, 4$.

| | | | |
|---|---|---|---|
| $C_{11} = -0.038150$ | $C_{12} = 0.000319$ | $C_{13} = 0.004707$ | $C_{14} = 0.049229$ |
| $C_{21} = 0.080202$ | $C_{22} = -0.055925$ | $C_{23} = 0.083370$ | $C_{24} = 0.148820$ |
| $C_{31} = 0.092563$ | $C_{32} = 0.092563$ | $C_{33} = -0.151266$ | $C_{34} = 0.092563$ |
| $C_{41} = 0.183898$ | $C_{42} = 0.183898$ | $C_{43} = 0.183898$ | $C_{44} = -0.341808$ |

Unfortunately, the transport theory data is not available to compare with for all critical reactors dimensions nor for their fluxes values.

The following subsections illustrate numerical results for three kinds of reactors geometry.

5.2.1. Spherical Reactor

The spherical reactor is studied by finding its critical radius of the reactors and distribution of four fluxes as well as their sum.

The critical radius $a_c$ for spherical reactor is calculated at ZF and EBC boundary conditions, respectively. The generated data are listed in Table 11.

**Table 11.** The spherical reactor critical radius $a_c$.

| BC | Classical Method | RPSM |
|----|------------------|------|
| ZF | 8.770 | 8.770 |
| EBC | 7.905 | 7.905 |

The values of the fluxes in the reactor is given in Table 12 and Figure 4. Table 12 provides four fluxes and their total flux values across the system, it is clear (in both Table 12 and Figure 4) that all fluxes decreases when the reactor radius increases and vanishes at the critical radius.

**Table 12.** The four groups fluxes and total flux in spherical reactor geometry.

| Flux | Method | $r/a_c$ | 0.0 | 0.25 | 0.50 | 0.75 | 1.0 |
|------|--------|---------|-----|------|------|------|-----|
| Group 1 | Classical | | 1.0000 | 0.918535 | 0.697804 | 0.400643 | 0.107646 |
| | RPSM | | 1.0000 | 0.918535 | 0.697804 | 0.400643 | 0.107646 |
| Group 2 | Classical | | 4.1716 | 3.83172 | 2.91093 | 1.67131 | 0.449052 |
| | RPSM | | 4.1716 | 3.83172 | 2.91093 | 1.67131 | 0.449052 |
| Group 3 | Classical | | 2.7361 | 2.5132 | 1.90926 | 1.0962 | 0.294531 |
| | RPSM | | 2.7361 | 2.5132 | 1.90926 | 1.0962 | 0.294531 |
| Group 4 | Classical | | 3.0931 | 2.84115 | 2.1584 | 1.23924 | 0.332964 |
| | RPSM | | 3.0931 | 2.84115 | 2.1584 | 1.23924 | 0.332964 |
| Total | Classical | | 11.0008 | 10.1046 | 7.67639 | 4.40739 | 1.18419 |
| | RPSM | | 11.0008 | 10.1046 | 7.67639 | 4.40739 | 1.18419 |

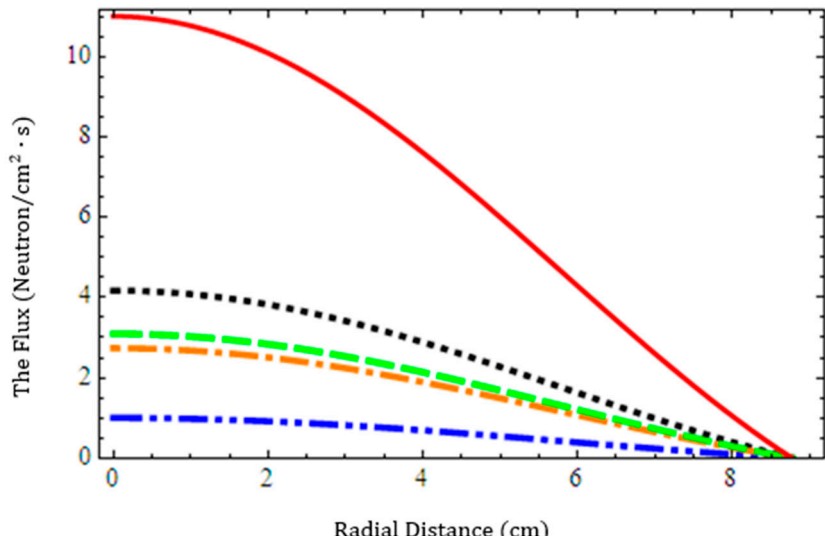

**Figure 4.** Four energy groups fluxes and total flux for a spherical reactor. Dotted-dotted dashed line: $\varnothing_1(r)$, Dotted line: $\varnothing_2(r)$, Dotted dashed line: $\varnothing_3(r)$, Dashed line: $\varnothing_4(r)$, Solid line: Total fluxes.

RPSM results (as well as for classical method) for multi-energy groups of neutrons system in spherical reactor assure and improve the calculations of that of two energy groups of neutrons.

5.2.2. Infinite Cylindrical Reactor

Now, RPSM is used in this section to calculate the infinite cylindrical reactor critical radius $a_c$, Equation (50) is implemented with ZF and EBC boundary conditions, the results are listed in Table 13.

**Table 13.** Cylindrical reactor critical radius for two boundary conditions.

| BC | Classical Method | RPSM |
|----|------------------|------|
| ZF | 6.71303 | 6.71303 |
| EBC | 5.84763 | 5.84763 |

The critical radius calculations using RPSM is confirmed using classical method calculation (there is no transport theory data to compare with as in the case of two energy groups).

The flux distribution of such cylindrical reactor is reported in Table 14 at which fluxes values across the system for different values of $r/a_c$ are listed. The RPSM values are compared with classical diffusion calculations [22–25]. Figure 5 illustrates fluxes distribution in this fissile system, the maximum value of the flux is at the axis of the cylinder ($r = 0$), it decreases towards the surface as implied by the symmetry of the system.

**Table 14.** The four groups fluxes and total flux in cylindrical reactor geometry.

| Flux | Method | $r/a_c$ | 0.0 | 0.25 | 0.50 | 0.75 | 1.0 |
|------|--------|---------|-----|------|------|------|-----|
| Group 1 | Classical | | 1.0000 | 0.9326 | 0.743977 | 0.471824 | 0.169557 |
| | RPSM | | 1.0000 | 0.9326 | 0.743977 | 0.471824 | 0.169557 |
| Group 2 | Classical | | 4.1716 | 3.8904 | 3.10354 | 1.96824 | 0.707317 |
| | RPSM | | 4.1716 | 3.8904 | 3.10354 | 1.96824 | 0.707317 |
| Group 3 | Classical | | 2.7361 | 2.55169 | 2.0356 | 1.29096 | 0.463925 |
| | RPSM | | 2.7361 | 2.55169 | 2.0356 | 1.29096 | 0.463925 |
| Group 4 | Classical | | 3.0931 | 2.88465 | 2.30122 | 1.45941 | 0.524462 |
| | RPSM | | 3.0931 | 2.88465 | 2.30122 | 1.45941 | 0.524462 |
| Total | Classical | | 11.0008 | 10.2593 | 8.18434 | 5.19044 | 1.86526 |
| | RPSM | | 11.0008 | 10.2593 | 8.18434 | 5.19044 | 1.86526 |

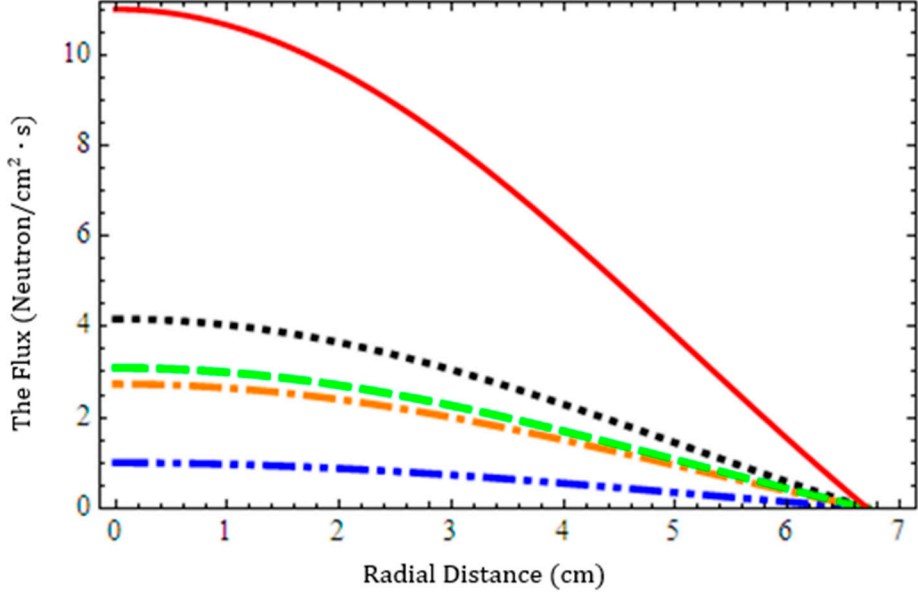

**Figure 5.** Four energy groups fluxes and total flux for a cylindrical reactor. Dotted-dotted dashed line: $\varnothing_1(r)$, Dotted line: $\varnothing_2(r)$, Dotted dashed line: $\varnothing_3(r)$, Dashed line: $\varnothing_4(r)$, Solid line: Total fluxes.

### 5.2.3. Slab Reactor

Computational analysis of four energy groups of neutrons at an infinite slab geometry is also performed. To calculate the critical dimension $a_c$ in slab reactors, Equation (51) implemented at ZF and EBC boundary conditions, the results are listed in Table 15.

**Table 15.** The slab reactor critical radius for both boundary conditions.

| BC | Classical Method | RPSM |
|---|---|---|
| ZF | 4.38485 | 4.38485 |
| EBC | 3.51945 | 3.51945 |

The four energy groups of neutron fluxes values in the slab reactor is reported in Table 16 and all fluxes are plotted in Figure 6; this figure illustrates the flux distribution of a slab fissile material. The RPSM values are compared to those obtained from classical diffusion calculations.

**Table 16.** The four groups fluxes and total flux in slab reactor geometry.

| Flux | Method | $r/a_c$ | 0.0 | 0.25 | 0.50 | 0.75 | 1.0 |
|---|---|---|---|---|---|---|---|
| Group 1 | Classical | | 1.0000 | 0.950736 | 0.807797 | 0.585267 | 0.305072 |
| | RPSM | | 1.0000 | 0.950736 | 0.807797 | 0.585267 | 0.305072 |
| Group 2 | Classical | | 4.1716 | 0.950736 | 0.807797 | 0.585267 | 0.305072 |
| | RPSM | | 4.1716 | 0.950736 | 0.807797 | 0.585267 | 0.305072 |
| Group 3 | Classical | | 2.7361 | 2.60131 | 2.21021 | 1.60135 | 0.834709 |
| | RPSM | | 2.7361 | 2.60131 | 2.21021 | 1.60135 | 0.834709 |
| Group 4 | Classical | | 3.0931 | 2.94075 | 2.49862 | 1.81031 | 0.943628 |
| | RPSM | | 3.0931 | 2.94075 | 2.49862 | 1.81031 | 0.943628 |
| Total | Classical | | 11.0008 | 10.4588 | 8.88641 | 6.4384 | 3.35604 |
| | RPSM | | 11.0008 | 10.4588 | 8.88641 | 6.4384 | 3.35604 |

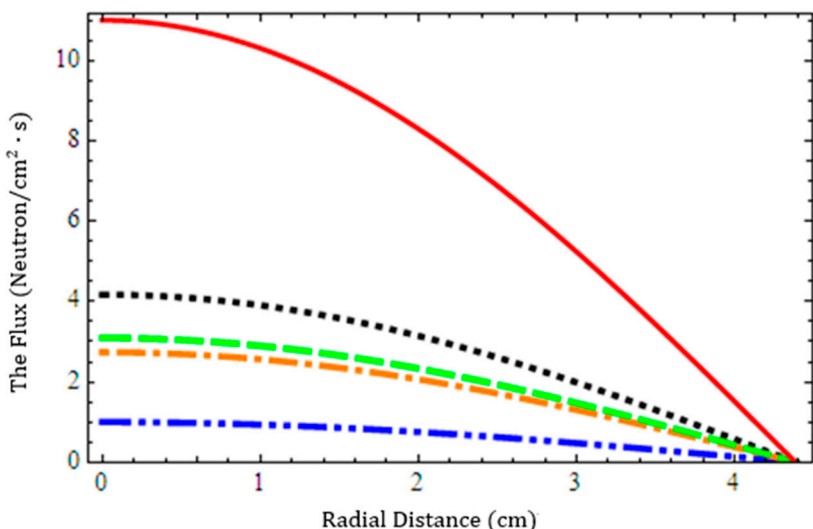

**Figure 6.** Four energy groups fluxes and total flux for a slab reactor. Dotted-dotted dashed line: $\varnothing_1(r)$, Dotted line: $\varnothing_2(r)$, Dotted dashed line: $\varnothing_3(r)$, Dashed line: $\varnothing_4(r)$, Solid line: Total fluxes.

## 6. Error Analysis

Since the transport theory data for four energy groups of neutrons is not available for comparison, we compared the results obtained in RPSM with the results obtained in the classical method only. In fact, both RPSM and classical methods provide us with series solutions. It is noted that the general terms in both series are different, but both solutions give us the same numerical results at the same

order of approximation. Therefore, we will make another comparison between the results obtained by RPSM and classical method by calculating the residual errors (Tables 17 and 18). For the sake of brevity, we will limit the comparison to one case, the spherical reactor for the four energy groups of neutron diffusion.

**Table 17.** The value of the 10th-residual error $(Res_i^{10}, i = 1, 2, 3, 4)$ for the results obtained by classical method.

| $r$ | $\mathbf{Res_1(r)}$ | $\mathbf{Res_2(r)}$ | $\mathbf{Res_3(r)}$ | $\mathbf{Res_4(r)}$ |
|---|---|---|---|---|
| 0.0 | 0 | 0 | 0 | 0 |
| 0.5 | 0 | $1.11022 \times 10^{-16}$ | $8.32667 \times 10^{-17}$ | $1.11022 \times 10^{-16}$ |
| 1.0 | $2.77556 \times 10^{-17}$ | $2.22045 \times 10^{-16}$ | $2.22045 \times 10^{-16}$ | $2.22045 \times 10^{-16}$ |
| 5.0 | $1.60982 \times 10^{-15}$ | $6.21725 \times 10^{-15}$ | $4.88498 \times 10^{-15}$ | $5.55112 \times 10^{-15}$ |
| 10. | $3.042870 \times 10^{-9}$ | $1.26935 \times 10^{-8}$ | $8.32559 \times 10^{-9}$ | $9.41198 \times 10^{-9}$ |
| 20. | $6.381350 \times 10^{-3}$ | $2.66202 \times 10^{-2}$ | $1.746 \times 10^{-2}$ | $1.97384 \times 10^{-2}$ |

**Table 18.** The value of the 10th-residual error $(Res_i^{10}, i = 1, 2, 3, 4)$ for the results obtained by RPSM.

| $r$ | $\mathbf{Res_1(r)}$ | $\mathbf{Res_2(r)}$ | $\mathbf{Res_3(r)}$ | $\mathbf{Res_4(r)}$ |
|---|---|---|---|---|
| 0.0 | 0 | 0 | 0 | 0 |
| 0.5 | 0 | 0 | $2.77556 \times 10^{-17}$ | $5.55112 \times 10^{-17}$ |
| 1.0 | 0 | $1.11022 \times 10^{-16}$ | 0 | $1.11022 \times 10^{-16}$ |
| 5.0 | $1.554310 \times 10^{-15}$ | $5.9952 \times 10^{-15}$ | $4.21885 \times 10^{-15}$ | $3.77476 \times 10^{-15}$ |
| 10. | $3.042866 \times 10^{-9}$ | $1.26935 \times 10^{-8}$ | $8.32559 \times 10^{-9}$ | $9.41198 \times 10^{-9}$ |
| 20. | $6.381350 \times 10^{-3}$ | $2.66202 \times 10^{-2}$ | $1.746 \times 10^{-2}$ | $1.97384 \times 10^{-2}$ |

The residual errors for the system in Equation (47) is defined as follows:

$$Res_i(r) = \left| r\varnothing_i''(r) + 2\varnothing_i'(r) + r(C_{i,1}\varnothing_1(r) + C_{i,2}\varnothing_2(r) + C_{i,3}\varnothing_3(r) + C_{i,4}\varnothing_4(r)) \right|, \; i = 1, 2, 3, 4, \quad (52)$$

and the $k$th-residual error is defined as:

$$Res_i^k(r) = \left| r\frac{d^2}{dr^2}\varnothing_i^k + 2\frac{d}{dr}\varnothing_i^k(r) + r\left(C_{i,1}\varnothing_1^k(r) + C_{i,2}\varnothing_2^k(r) + C_{i,3}\varnothing_3^k(r) + C_{i,4}\varnothing_4^k(r)\right) \right|, \; i = 1, 2, 3, 4, \quad (53)$$

where $\varnothing_i^k$ is the $k$th-approximation of $\varnothing_i$, $i = 1, 2, 3, 4$.

## 7. Conclusions

RPSM is applied in solving multi-energy groups of a neutron diffusion equation for different reactor geometries. This method successfully analyzes both two and four energy groups of neutron diffusion systems numerically. The flux distributions of multi-energy groups of a neutron diffusion system are presented and the determination of each nuclear reactor critical radius is illustrated. The calculated data is compatible with other compared methods such as the classical method and the available transport theory data. The efficiency of RPSM appears in introducing it as a new approximate method which can reproduce nuclear reactor fluxes directly, without using additional methods, the numerical results can be reached easily too. Moreover, RPSM, like other analytical methods, provided a series solution for multiple energy groups of neutron diffusion equations and gave us the same results obtained in classical and homotopy perturbation methods at the same order of approximation because we deal with linear equations. Despite all this, the advantage of this method is the ease, and speed of the solution in RPSM. We expect that the RPSM application on nonlinear equation models will prove to be more accurate than other analytical methods.

**Author Contributions:** Conceptualization, M.S.; Data curation, M.N.; Methodology, A.E.-A.; Project administration, M.S.; Software, M.N.; Writing—original draft, M.S.

**Funding:** The financial support provided by the deanship of scientific research at Prince Sattam Bin Abdulaziz University is gratefully acknowledged.

**Acknowledgments:** The financial support of the project (Using New Mathematical Methods in Solving Neutron Diffusion Equations in Bare and Reflected Nuclear Reactors) funded by the Deanship of Scientific Research at Prince Sattam Bin Abdulaziz University within the Specialized Research Grant program and under contract number 2017/01/7676 is gratefully acknowledged.

**Conflicts of Interest:** The authors declare no conflict of interest.

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
