# Peer review of "Analytical Solution for Multi-Energy Groups of Neutron Diffusion Equations by a Residual Power Series Method"

_mathematics, doi:10.3390/math7070633_

Round 1

Reviewer 1 Report

Some of my comments are addressed, however I still would like to request the authors to address my remarks listed below,

1. I have some concerns about the comparison between classical method and proposed RPSM results. These two approaches give exactly same results and should we expect this be the case? Does the accuracy of RPSM depend on the order of approximation? A necessary remark on this matter would be necessary. 

2. In addition, the efficiency comparison of classical method, RPSM and transport theory is still not shown at all. If it is not possible to compare these three approaches, authors at least should compare the efficiency of RPSM in terms of different energy groups and approximation order. 

Author Response

 Dear Dr. (the reviewer in mathematics journal),

I would like to thank you for your useful suggestions and comment in our paper which titled  "Analytical solution for multi energy groups of neutrons diffusion

equations by residual power series method" .

We have made all the comments and corrections as suggested.

The response of comment one is:

In fact, both RPSM and classical methods provide us with series solutions, and it is remarkable that the overall the general terms in both series are different but both solutions give us the same numerical results at the same order of approximation. Also, as any series solution, increasing the order of approximation increases the accuracy of the solution.  (Please see the paragraphs (16.3 through 16.9) and (8.17 through 8.20) in the revised version).

The response of comment two is:

In response, we added new section "Error Analysis" showing the efficiency of the RPSM. (Please see the revised version, Section 6)

note the revised version of the manuscript  is attached 

Reviewer 2 Report

This is an interesting and new approach to multigroup diffusion equation. Typical nomenclature is "Multiple energy group" or "Multigroup" not "Multi energy groups", but that is author's preference. Subsection headings should use a consistent case. I recommend publication pending typographical changes. 

Below are some minor comments and suggestions.  Each is labeled as [page].[line]. 

1.43 "is product" -> "is a product"

2.33 "where ???is known" - missing space after Cii

2.33 Missing "," before "and" 

3.19 through 4.5 Sometimes there is a space after "Step N:", sometimes not. It should be consistent. 

5.11 and 5.13 "Ti.2k" shoudl be "Ti,2k"

7.16 missing space before "and"

7.18 "Ti.0" should be "Ti,0"

7.19 likewise, "Ti.X" should be "Ti,X"

8.2, 8.3, 8.22, 8.23, 8.24 "Ti.2k" should be "Ti,2k"

8.8 "vanish" -> "vanishes"

8.9 "the critical dimension"

8.16 "Two ... "

9.12 "Table 2: The critical radius ?? of a 93 % enriched Uranium spherical reactor"

9.14 its (or it is)  not it's

9.18 How are the fluxes normalized?

9.19 "Table 3: The normalized thermal, fast, and total fluxes in a spherical geometry." 

9.19 I would remove the lines for transport method in Table 5 if there are no results. 

8.26 "numerical results classical solution" -> "numerical results with a classical solution"

9.2 missing space before "and"

9.2 table -> Table

9.3 missing space after "coefficients"

9.5 "Following subsections illustrate numerical results for each kind of reactors geometry." -> "Following subsections show numerical results for each reactor geometry."

10.0, 11.0, 12.0, 14.4,16.1 Missing units on y-axis of Figures 1, 2, 3, 4, 5

10.2 Figure 1

10.2 "it is" not it's

10.12 I would remove the column for transport method in Table 4 if there are no results. 

10.17 Figure 2

10.18 Table 5

10.21 missing "," before "and"

10.21 I would remove the lines for transport method in Table 5 if there are no results. 

12.4  Multi..

12.10 missing "," before "and"

12.13, 12.14, 12.15 "Ti.2k" should be "Ti,2k"

12.18 numerical results "of the" classical solution

13.1 missing "," before "and"

13.3 Correct units are: MeV, keV, and eV

13.5 missing space before "and"

13.5 Table 8

13.7 missing space after "coefficients"

13.23 "it is" not it's

13.23 "all fluxes decrease" not "all fluxes are decreases"

15.13 Table 14 seems to be missing numbers for RPSM results at r/ac > 0.

16.2-3 Suggestion to reformulate for clarity: "There is no transport theory data to compare with in four energy groups of neutrons in the slab reactor. However, the generalization of two energy groups to four energy group can be considered as an achievement."

16.8 "The flux distributions..."

16.9 remove "too"

16.10 ".. such as the classical method and the available .."

16.18 References should use a consistent format, sometimes there is a comma before page numbers, missing spaces before (YEAR) etc. 

17.37 Reference [25] was cut off. 

Author Response

Dear Dr. (the reviewer in mathematics journal),

I would like to thank you for your useful suggestions and comment in our paper which titled "Analytical solution for multi energy groups of neutrons diffusion

equations by residual power series method" .

We have made all the comments and corrections as suggested.

Note: All improvements and modifications to the revised version are highlighted in green color.

All typos have been corrected in green.

All comments and suggestions have been considered and taken into consideration.

(Please see the revised version)

note the revised manuscript is attached

This manuscript is a resubmission of an earlier submission. The following is a list of the peer review reports and author responses from that submission.

Round 1

Reviewer 1 Report

- The authors aimed at solving analytically the multigroup neutron diffusion equation by residual power series method for three homogeneous geometries: a slab, a cylinder and a sphere. The results are presented for two energy groups. The background of this paper is interesting but more applicability is needed. The geometries presented are simple cases and they are very studied. A more challenging problem increases the quality of the paper with more energy groups.

- The results are compared with other solvers. First, with a classical method (unknown, neither described nor mentioned) where the numerical results are exactly the same that the obtained ones with the classical method, but the authors do not show the efficiency of the method proposed. The efficiency of this method must be shown, analytically or numerically. Then, numerical results are compared with transport theory data, but only for a few cases. 

- The description of the problem must be defined with more detail, indicating, for instance, what represents $\Sigma_f$, $\Sigma_{\gamma}$,... etc Moreover, it is needed to define the type of boundary conditions that then, they are called in the numerical results section.

- I recommend also described the RPSM by using the variables of the neutron diffusion equations in order to facilitate their comprehension and clarity.

- There are many Tables that they are not commented in the text (only presented). 

- Finally, I recommend improving the English, mainly in nuclear technical terms as: 'two-group neutrons' by 'two energy groups of neutrons', 'flow of neutrons' by 'neutron flux', 'critical radii' etc

- There are also other writing mistakes that have to be corrected. 

- Figure 2 does not observe well in the manuscript.

Reviewer 2 Report

This manuscript develops analytical solution for multi-group neutron diffusion equations using residual power series method. The obtained solution has good agreement with the solution obtained by alternative approaches. The manuscript fits within the scope of Mathematics. My overall impression of the manuscript is positive. With minor revisions, in my opinion, the manuscript is suitable for publication. In order to further improve the manuscript, I would like to request the authors to address my remarks listed below:

1 Authors conclude that their RPSM approach is efficient compared to classical method and transport theory. However, no numerical example is presented to support this argument. A fair comparison of the efficiency for the three methods seems to be necessary. 

2 What is order "k" used for those numerical examples? Some comments are necessary on the choice of k.

3 The authors performs numerical studies on two-group neutron diffusion equations. What are the advantage and disadvantage of RPSM for multi-group systems compared to classical methods.